# Volumetric Neural Human for Robust Pose Optimization via Analysis-by-synthesis

**Pengliang Ji**
Beihang University
jpl1723@buaa.edu.cn

**Angtian Wang**
Johns Hopkins University
angtianwang@jhu.edu

**Yi Zhang**
Johns Hopkins University
yzh@jhu.edu

**Adam Kortylewski**
Max Planck Instutite for Informatics
akortyle@mpi-inf.mpg.de

**Alan Yuille**
Johns Hopkins University
ayuille1@jhu.edu

## Abstract

Regression-based approaches dominate the field of 3D human pose estimation, because of their quick fitting to distribution in a data-driven way. However, in this work we find the regression-based methods lack robustness under out-of-distribution, i.e. partial occlusion, due to its heavy dependence on the quality of prediction of 2D keypoints which are sensitive to partial occlusions. Inspired by the neural mesh models for object pose estimation, i.e. meshes combined with neural features, we introduce a human pose optimization approach via render-and-compare neural features. On the other hand, the volume rendering technical demonstrate better representation with accurate gradients for reasoning occlusions. In this work, we develop a volumetric human representation and a robust inference pipeline via volume rendering with gradient-based optimizations, which synthesize neural features during inference while gradually updating the human pose via maximizing the feature similarities. Experiments on 3DPW show ours better robustness to partial occlusion with competitive performance on unoccluded cases.

## 1 Introduction

3D human mesh reconstruction via pose estimation from a single RGB image shows remarkable significance in real-world applications, e.g., VR/AR and robotics. The regression-based approach can quickly fit the 3D pose parameters of human body models such as SMPL Loper et al. (2015) in a data-driven way. However, learning directly from large amounts of training data, the approach is vulnerable to out-of-distribution, i.e. partial occlusion. This lack of robustness can have severe consequences in real-world applications and therefore is urgently needed to be addressed.

Recent works Kocabas et al. (2021); Wang et al. (2020); Cheng et al. (2019) enhance the robustness of human pose estimation by following two steps: First, detect dense 2D keypoints of the human body by deep neural networks. Then, regress towards the feature of predicted non-occluded 2D keypoints with additional knowledge for robustness. However, Their robustness is limited because it relies heavily on the quality of predicted 2D keypoints, which are sensitive to partial occlusions. On the other hand, the direct use of a downsampled mesh model to represent the human body in recent works leads to a failure of reasoning occlusion. Because the naive mesh lacks human-related insight, it is particularly unsuitable for representing human arms and legs, which are thin and long in shape.

Given the success of volume rendering on other tasks such as 3D reconstruction Mildenhall et al. (2020), we exploit insights to apply volume rendering, which demonstrates better representation with

4th Workshop on Shared Visual Representations in Human and Machine Visual Intelligence (SVRHM) at the Neural Information Processing Systems (NeurIPS) conference 2022. New Orleans.

accurate gradients for reasoning occlusions, on a new task human pose estimation with a volume neural renderer Wang et al. (2022). On the other hand, recent works have shown the power of neural analysis-by-synthesis in performing visual recognition tasks such as image classification Kortylewski et al. (2020) and object pose estimation Wang et al. (2021) with render-and-compare. Therefore, we want to eliminate the robustness dependence on 2D keypoints by replacing them with rendering kernels in volume space which make up a volumetric human representation, and for the first time show the volume rendering with render-and-compare in developing robustness for human pose estimation.

In this paper, we introduce a volumetric neural human for robust pose optimization that is highly robust to occlusion. Our key idea is to extend SMPL human body model to a volumetric human representation with a texture map which consists of a learned neural feature to perform model fitting on the level of neural network features with volume-renderer via render-and-compare. First, we propose a volumetric human representation designed with human-relevant insight to better represent the human body, and a corresponding texture map capturing the local information of the body parts. Then, we introduce a feature extractor with contrastive learning, to learn local pose information of the human body parts with invariance to instance-specific details (such as the color of clothes), and a generative model of feature activations at every rendering kernel of the volumetric human, to capture the remaining variability in the feature activations of humans in the training data. During model inference, we optimize the human pose by maximizing the likelihood of the target feature map under our generative model using gradient-based optimization. By render-and-compare, we predict human pose parameters with high robustness and avoid ambiguity, benefiting from volumetric human representation and volume rendering. We find that our method can resolve common errors of regression-based methods under occlusion.

We evaluate our method at human pose estimation on the 3DPW benchmark von Marcard et al. (2018a) and its occluded versionKocabas et al. (2021). Experiment on 3DPW shows ours better robustness to partial occlusion than the state-of-the-art regression-based approach. More expressively, while bringing robustness to deal with the occluded data, our method keeps predicting non-occluded data accurately without introducing bias commonly brought by optimization towards out-distribution. In summary, our main contributions are:

1. We reveal the simple mesh model widely used in existing methods without human-relevant insights does not effectively represent humans, and design a volumetric neural human representation consisting of 3D Gaussian ellipsoids kernels that characterize and fit each component of the human body distinctly and adaptively.

2. We propose a generative model and contrastive learning framework with better occlusion reasoning and low ambiguity, to achieve a robust human pose optimization by fitting at the feature level via analysis-by-synthesis with volume rendering.

3. We conduct experiments on a popular 3D pose estimation benchmark featuring occluded bodies and outperform prior arts with better robustness, for the first time showing an impressive potential of volume rendering in improving robustness for human pose estimation.

## 2 Method

In the following, we introduce our model by first describing the volumetric neural human representation, and then our full neural human body fitting framework for robust human pose optimization.

### 2.1 Volumetric Neural Human

A straightforward way of extending an analysis-by-synthesis generative model to 3D human pose estimation, is to replace the mesh with a parametric human body model $M(\theta, \beta)$, such as the SMPL model Loper et al. (2015). It takes as input the pose parameters $\theta \in \mathbb{R}^{24 \times 3}$ and the shape parameters $\beta \in \mathbb{R}^{10}$ to produce the body mesh $M \in \mathbb{R}^{N \times 3}$, where $N = 6890$. Due to the limited resolution of feature maps, prior works directly use downsampled mesh as a representation without a human-relevant strategy, as was done in the object pose estimation task, leading to a loss of information.

Inspired by the human intuition to judge body pose from global to local and the fact that the rotation of body components is always joint-centered, our insight is to place Gaussian ellipsoids kernels, which adaptively adjust their shape according to the belonging body part, on the human skeleton and joints in volume space. It takes into account detailed body components not covered by the mesh

representation, and brings potential information at the part-level by treating vertices belonging to the same part equally. During rendering, the human volume $\rho$ are reconstructed with a sum of kernels, and the observed color $C(r)$ along a ray $r(t) = (x(t), y(t), z(t))$ are computed with a volume rendererWang et al. (2022). Please refer to Appendix.A for more details of design and rendering.

Based on that, given the volumetric neural human $\mathfrak{N}_y = \{V, \Phi\}$ consists of a set of Gaussian ellipsoids kernels $V \in \mathbb{R}^{N \times 3}$ with feature vectors $\Phi = \{\phi_r \in \mathbb{R}^D\}$, human pose estimation with better occlusion reasoning are done by comparing the rendered feature map obtained via ray-tracing with the feature map extracted directly from images, which will be introduced in detail next.

## 2.2 Generative Model of Volumetric Neural Human Textures

Following and extending Neural Mesh Models (NMMs)Wang et al. (2021) via analysis-by-synthesis, we define a probabilistic generative model $p(F|\mathfrak{N}_y)$ of the real-valued feature activations $F$ of an object class $y$. The Feature map can be rendered from the volumetric neural human $\mathfrak{N} = \{V, \Phi\}$ using a differentiable volume renderer Wang et al. (2022), i.e. $\bar{F}(\Pi) = \mathfrak{R}(\mathfrak{N}, \Pi) \in \mathbb{R}^{H \times W \times D}$, where $\Pi$ are the camera parameters. Then, following related work on robust inference with generative models Egger et al. (2018), the model likelihood can be made with robustness to occlusion:

$$p(F|\mathfrak{N}, \Pi, B, z_i) = \prod_{i \in \mathcal{FG}} \left[ p(f_i|\mathfrak{N})p(z_i{=}1) \right]^{z_i} \left[ p(f_i|B)p(z_i{=}0) \right]^{(1-z_i)} \prod_{i' \in \mathcal{BG}} p(f_{i'}|B), \quad (1)$$

where $z_i \in \{0, 1\}$ is a binary variable and $p(z_i{=}1)$ and $p(z_i{=}0)$ are the prior probabilities of the respective values. The variable $z_i$ allows the background model $p(f_i|B)$ to explain those locations in the feature map $F$ that is in the foreground region $\mathcal{FG}$ but which the foreground model $(f_i|\mathfrak{N})$ cannot explain well, presumably due to partial occlusion. To reduce clutter in the remaining paper, we will omit the occlusion variable in the coming equations, but note that we are using a robust likelihood during inference. Our model pipeline and derivation of the formula are introduced in Appendix.A.

**MLE Training.** We train the parameters $\Phi$ of the generative probabilistic model through maximum likelihood estimation (MLE) by minimizing the negative log-likelihood of the feature representations over the whole training set, where the correspondence between feature vectors $f_i$ and kernels $r$ is computed using the ground-truth pose and camera parameters. After deriving and simplifying the negative log-likelihood loss function, we use below mean squared error between the volume kernel features and the target feature map as the loss function:

$$\mathcal{L}_{NLL}(F, \mathfrak{N}, \Pi, B) = \frac{1}{2} \sum_{i \in \mathcal{FG}} \|f_i - \phi_r\|^2 + \frac{1}{2} \sum_{i' \in \mathcal{BG}} \|f_{i'} - b\|^2 + C, \quad (2)$$

where $C$ is a constant scalar. This loss function can be optimized via gradient descent using a differentiable volume renderer, e.g. Wang et al. (2022); Keselman & Hebert (2022) by ray tracing.

**Contrastive Learning of the UNet Feature Extractor.** To avoid local optima in the reconstruction loss, we train our feature extractor by contrastive learning to learn features and hence become aware of the local 3D pose of the limbs. We achieve this by optimizing a contrastive loss with three terms:

$$\mathcal{L}_{Kernel}(F, \mathcal{FG}) = -\sum_{i \in \mathcal{FG}} \sum_{i' \in \mathcal{FG} \setminus \{i\}} \|f_i - f_{i'}\|^2 \quad (3)$$

$$\mathcal{L}_{BG}(F, \mathcal{FG}, \mathcal{BG}) = -\sum_{i \in \mathcal{FG}} \sum_{j \in \mathcal{BG}} \|f_i - f_j\|^2 \quad (4)$$

where $\mathcal{L}_{Kernel}$ encourages features of different kernels to be distinct from each other. $\mathcal{L}_{BG}$ encourages features on the human to be distinct from those in the background. We optimize those losses jointly $\mathcal{L}_{contrast} = \mathcal{L}_{Kernel} + \mathcal{L}_{BG}$ in a contrastive learning framework. Our full model is trained by optimizing the overall training loss $\mathcal{L}_{train} = \mathcal{L}_{NLL} + \mathcal{L}_{contrast}$, where we need to optimize the parameters of the UNet feature extractor $\zeta$ and the parameters of the generative model $\Phi$ jointly.

**Optimization with Multi-task Integration.** Once the generative model is trained with a general ability to represent humans wisely, we perform 3D human pose estimation for each instance by minimizing Equation 2, which is also used for training. However, this is non-convex optimization that can easily converge to local minima if the initialization is not good enough. Motivated by existing optimization-based methods Pavlakos et al. (2019); Bogo et al. (2016), we introduce auxiliary losses to guide the optimization process:

$$\mathcal{L}_{reproj}(\hat{V}_{2D}, \mathfrak{N}, \Pi) = \sum_i \|\hat{V}_{2D}^i - \Pi(W_i M)\|^2 + \sum_r o_r \|\hat{v}_r - \Pi(V_r)\|^2 \quad (5)$$

where $\hat{V}_{2D}$ is the kernel locations detected by an off-the-shelf detector Cao et al. (2017) and $W_i \in \mathbb{R}^{1 \times N}$ is a pretrained linear regressor. $V_{i'} \in M$ denotes a kernel of which the visibility is $o_r$ and the maximum-likelihood detection of the 2D location is $\hat{v}_r = argmax_{(h,w)}\|f_{h,w} - \phi_r\|^2$. We also involve part segmentation loss $L_{partseg}$ and prior loss $L_{prior}$ ensuring the 3D pose is valid in our inference. The final objective at inference time is:

$$\mathcal{L}_{inference}(F_j, \mathfrak{N}, \Pi, B) = \mathcal{L}_{NLL}(F_j, \mathfrak{N}, \Pi, B) + \mathcal{L}_{reproj}(\hat{V}_{2D}, \mathfrak{N}, \Pi) \qquad (6)$$
$$+ \mathcal{L}_{partseg}(\hat{P}, \mathfrak{N}, \Pi) + \mathcal{L}_{prior}(\mathfrak{N}).$$

where $\hat{P}$ is the observation of part segmentation, please refer to Appendix.A for more details.

## 3 Experiments

### 3.1 Training Setup and Datasets

**Training.** We train all on **COCO** Lin et al. (2014) datasets for 175K iterations at the same setting.

**Evaluation.** For evaluation, we use the in-the-wild dataset **3DPW** von Marcard et al. (2018b) to measure the robustness and generalization of our method. As the main evaluation metrics, mean per joint position error (MPJPE), Procrustes-aligned mean per joint position error (PA-MPJPE).

**Adversarial Occlusion Robustness Evaluation.** Instead of using a gray occlusion patch, we use textured patches generated by randomly cropping texture maps which is more challenging. Two different patch sizes are used: $40 \times 40$ and $80 \times 80$ for a $224 \times 224$ image, denoted as Occ@40 and Occ@80 respectively. More details about the setup and implementation are expanded in Appendix.A.

### 3.2 Performance Evaluation

**Comparison to State-of-the-art.** We evaluate the occlusion robustness of three SoTA regression-based methods on 3DPW-AdvOcc. As shown in Table 1, our volumetric neural human is more robust to occlusion than the state-of-the-art robust method Kocabas et al. (2021). While in the non-occluded setting, our method achieves comparable or better performance. We also use 2D Percentage of Correct Keypoints with head length threshold (PCKh) as our metric to show much more robustness to predict 2D keypoints, which existing methods for robustness heavily rely on. Note that we compared with the state-of-the-art works that have essential improvements, so some most recent works such as Li et al. (2022) which introduces performance improvements at the data level are excluded.

Table 1: **Performance on 3DPW and 3DPW-AdvOcc.** Ours outperforms state-of-the-art regression-based methods on 3DPW-AdvOcc while being on par or better on 3DPW. Evaluation metrics reported: MPJPE (mm, ↓ the better), PA-MPJPE (mm, ↓ the better), and PCKh (%, ↑ the better).

| Method | 3DPW | | | 3DPW-AdvOcc@40 | | | 3DPW-AdvOcc@80 | | |
|---|---|---|---|---|---|---|---|---|---|
| | MPJPE | PAMPJPE | PCKh | MPJPE | PAMPJPE | PCKh | MPJPE | PAMPJPE | PCKh |
| SPIN Kolotouros et al. (2019) | 95.08 | 57.40 | 91.84 | 113.89 | 69.36 | 85.84 | 155.98 | 89.20 | 71.92 |
| HMR-EFT Joo et al. (2021) | 89.88 | 53.51 | 92.55 | 108.27 | 66.15 | 87.35 | 142.74 | 82.62 | 78.81 |
| PARE Kocabas et al. (2021) | 81.44 | **50.92** | 92.48 | 93.18 | 61.31 | 89.15 | 117.67 | 72.90 | 84.27 |
| Ours | **81.43** | 51.05 | **94.28** | **91.92** | **58.42** | **91.76** | **114.74** | **68.70** | **86.85** |

**Ablation Studies.** We verify the usefulness of the volumetric human representation described in Sec. 2.1 and loss functions for inference with results shown in Table 2. 2D keypoint reprojection loss contributes a lot because directly optimizing the $NLL$ loss does not generate good results as the initialization can be very far away from the global optima when the subject is under heavy occlusion. Conversely, only fitting the 2D losses gives inferior results either because of lacking 3D awareness.

Table 2: **Ablation Studies on 3DPW and 3DPW-AdvOcc.** Ours outperforms state-of-the-art regression-based methods on 3DPW-AdvOcc while being on par or better on 3DPW. Evaluation metrics reported: MPJPE (mm, ↓ the better), PA-MPJPE (mm, ↓ the better).

| Method | 3DPW | | 3DPW-AdvOcc@40 | | 3DPW-AdvOcc@80 | |
|---|---|---|---|---|---|---|
| | MPJPE | PA-MPJPE | MPJPE | PA-MPJPE | MPJPE | PA-MPJPE |
| Ours w/o VolHuman & Keyp. 2D & NLL | 99.47 | 65.39 | 104.68 | 72.13 | 133.94 | 73.83 |
| Ours w/o VolHuman & Keyp. 2D | 89.91 | 59.10 | 99.87 | 66.08 | 125.81 | 71.39 |
| Ours w/o VolHuman | 83.66 | 52.35 | 93.32 | 59.92 | 118.78 | 69.87 |
| Ours | **81.43** | **51.05** | **91.92** | **58.42** | **114.74** | **68.70** |

## 4    Conclusion and Future Work

In this work, we introduce volumetric neural human, an optimization system for human pose estimation that is accurate and highly robust to occlusion. First, we design a volumetric human representation consisting of 3D Gaussian ellipsoids kernels that fit each component of the human body adaptively, with a corresponding neural texture map to capture pose information. By proposing a generative model and contrastive learning framework with better occlusion reasoning, we achieve robust human pose optimization via analysis-by-synthesis with volume rendering. Experiments on a challenging benchmark dataset show our optimization has better robustness to partial occlusion and still be accurate on unoccluded cases. We also verify the usefulness of each component and the potential of volume rendering in human pose estimation. In the future, we will further improve volumetric neural humans by involving more human-relevant insight, such as view-point information, to overcome ambiguity, and apply volume rendering on more tasks to develop their robustness.

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

# A  Appendix

## A.1  Overview of our System

In this work, we propose the volumetric neural human for robust pose optimization via analysis-by-synthesis. Given the designed volumetric neural human representation $\mathfrak{N} = \{V, \Phi\}$ consists of a set of Gaussian ellipsoids kernels $V \in \mathbb{R}^{N \times 3}$ on human body $M(\theta, \beta)$ with feature vectors $\Phi = \{\phi_r \in \mathbb{R}^D\}$, we conduct training and inference for robust human pose estimation. During training, we train the parameters $\Phi$ through maximum likelihood estimation (MLE) with a volume renderer to make the generative model better recognize the volumetric human. On the other hand, we train the feature extractor U-Net through contrastive learning to extract features distinctly to avoid local optima. After that, we conduct human pose optimization through the trained model with better occlusion reasoning in the differentiable pipeline with the aid of optimization loss.

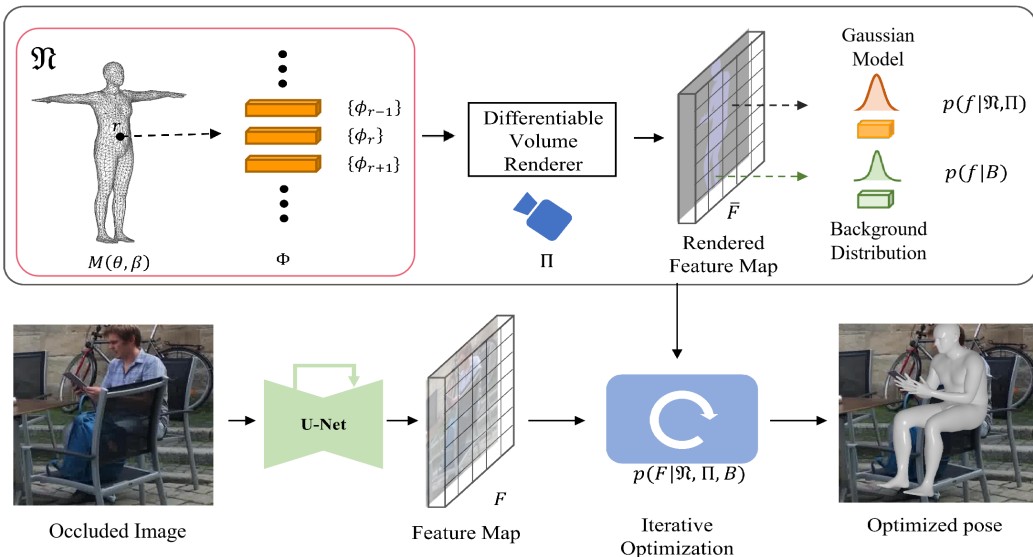

Figure 1: Overview of our system. we perform robust human pose optimization by fitting a generative model of deep feature vectors to the feature map $F$ extracted by a U-Net. Our model consists of a parametric volumetric neural human body $M(\theta, \beta)$ with neural texture map $\Phi$ which stores a set of features for each rendering kernel. After rendering the volumetric neural human to $\bar{F}$ with volume renderer, feature distributions for foreground pixels are modeled by a Gaussian model and background pixels are modeled by a background model. With the likelihood of $F$ from training, robust pose estimation is done by optimizing the negative log-likelihood loss w.r.t. the pose parameters and the camera parameters.

## A.2  Optimization Loss

Here we develop more details of loss functions for inference. Besides 2D reprojection loss of detected kernels, We consider a part segmentation loss to introduce more information for occlusion reasoning. It makes the part segmentation $P_i$ of each kernel $i$ optimized towards ground truth $\hat{P}_i$:

$$\mathcal{L}_{partseg}(\hat{P}, \mathfrak{N}, \Pi) = \sum_i CrossEntropy(\hat{P}_i, P_i) \tag{7}$$

where the observation of part segmentation $\hat{P}$ is obtained using Kocabas et al. (2021). We also use a 3D pose prior Pavlakos et al. (2019) to ensure the 3D pose is valid:

$$\mathcal{L}_{prior}(\mathfrak{N}) = \mathcal{N}(0, I)(E_{vae}(\theta)) \tag{8}$$

where $E_{vae}$ is the encoder of a pretrained variational autoencoder (VAE) which encodes the pose parameter to a 32-dimensional latent variable that is normally distributed.

### A.3 Volumetric Neural Humans

As we mentioned before, the downsampled mesh, which is widely used to represent objects, now is directly used for human pose estimation by many works, leading to bad robustness reasoning. Specifically, this simple representation can not represent the human body well, especially for the limbs of humans, where the vertice of mesh distribution is the most unevenly as shown in Fig 4. Besides, each vertex of mesh representation is considered equal in weight because of their lacking characters in size. This shortage of human-relevant insight representation leaves the rendering process without critical information and out of control.

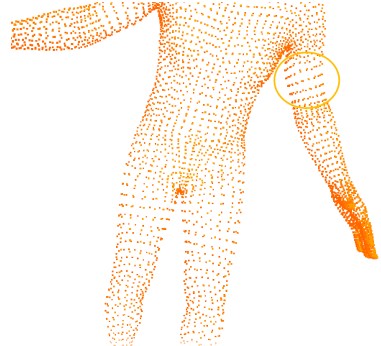

Figure 2: Unevenly distribution of vertices of mesh representation. With a shortage of human-relevant insight, this representation is especially not suitable for representing thin-shaped human limbs.

Instead, the volumetric neural human representation not only avoids the dependence on 2D keypoints with volume rendering, but also involves more knowledge for occlusions reasoning by representing instinct information from vertex level to parts level of the body model. In this section, we will develop more details of our volumetric representation designed for human, including geometry design, quantitative comparisons and rendering process.

#### A.3.1 Geometry Design

The geometric design of volumetric human representation refers to the arrangement of rendering kernels that take into account the shape of the human body, like the vertices in the mesh representation. It includes location, size and number of rendering kernels.

Concretely, the volumetric neural human representation consists of $K$ ellipsoids gaussian kernels with different sizes and shapes controlled by a parameter $\Sigma$, whose formula will be given in the following subsection A.3.3. As shown in Fig 3, by easily changing the scale and lineage of $\Sigma$ based on the shape of the human part to which it belongs in our implementation, $K$ rendering kernels can adaptively fit every shape of humans.

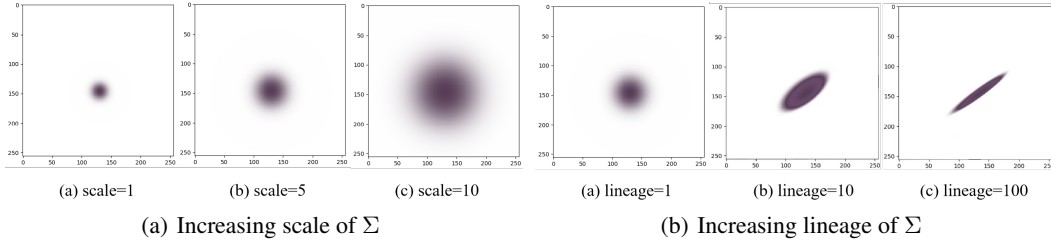

(a) scale=1    (b) scale=5    (c) scale=10      (a) lineage=1    (b) lineage=10    (c) lineage=100

(a) Increasing scale of $\Sigma$          (b) Increasing lineage of $\Sigma$

Figure 3: Adjustment of the parameters $\Sigma$ of the kernel to fit the shape of the human body. Each rendering kernel automatically adjust the scale and lineage of $\Sigma$ to achieve the best way to represent the human body in the volumetric level.

Based on rendering kernels adaptively fitting to the body part it locate at, inspired by the principle of human structure, they are placed uniformly on the human skeleton in the number of $N$, as well as on

Table 3: Comparisions of two representations' kernel

| Representation | Location | Amounts |
|---|---|---|
| Downsampled mesh | Vertice | 858 |
| volumetric human | Midpoints of skeleton | 439 |

the J joints in the SMPL model. Fig 4 shows their arrangement in a human body. In this way, the human representation in volume space not only takes into account those detailed body components that cannot be covered by the mesh representation, but also brings more potential information, such as a part-level segmentation obtained by treating vertices belonging to the same part equally. In our implementation, $L = 18$, $N = 23$, $J = 25$, $K = 439$.

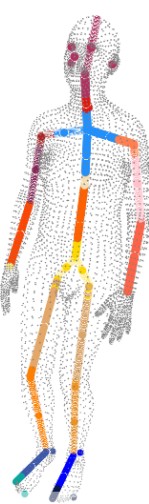

Figure 4: Layout position of kernels to render in a human body. They are placed uniformly on the human skeleton and the joints in the SMPL model.

### A.3.2 Comparisons of Two Representations

Here we give comparisons of rendering differences between volumetric representation and downsampled mesh representation. In Table 3 we show the difference of kernels to render in two representations. For the downsampled mesh, rendering kernels are put on each vertex of downsampled mesh with neural features to represent the human body. It needs more than 800 vertices to validly cover the human body. For a volumetric human, rendering kernels are mainly put on the midpoints of the skeleton so that only half the number of vertices used by the mesh one is needed to effectively represent humans at the volumetric level. Based on that, we give comparisons of training memory(Mbytes/GPU) and speed(images/s/GPU) during the rendering of two representations, as shown in Fig 5(a). Rendering in volumetric space with volumetric human representation a 30% reduction in computational memory and a 36% increase in speed.

We demonstrate the visualization of kernels for volumetric neural humans from different views with a reference to the mesh model. To simplify here we set $N = 1$, i.e. each body part is represented by one kernel. The qualitative results is shown in Fig 6.

### A.3.3 Rendering Process

Here we give the core part of the rendering process for our volumetric human representation with a volume renderer. For volume rendering, as Mildenhall et al. (2020) introduce, objects are represented using continuous volume density functions $\rho(x, y, z)$ with emitted color $c(x, y, z) = (r, g, b)$. The observed color $C(r)$ along a ray $r(t) = (x(t), y(t), z(t))$ are computed by:

$$C(r) = \int_{t_n}^{t_f} T(t)\rho(\mathbf{r}(t))\mathbf{c}(\mathbf{r}(t))dt, \; where \; T(t) = exp\left(-\gamma \int_{t_n}^{t} \rho(\mathbf{r}(s))ds\right) \tag{9}$$

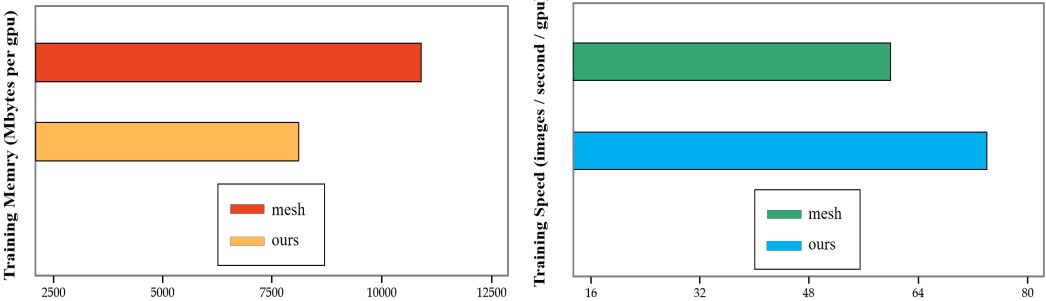

(a) The comparison of training memory (Mbytes/GPU) (b) The comparison of training speed (image/s/GPU)

Figure 5: The Comparisons of training memory and speed during rendering of downsampled mesh and volumetric human. Rendering in volume space, the volume neural human achieves better computational speed and less memory usage.

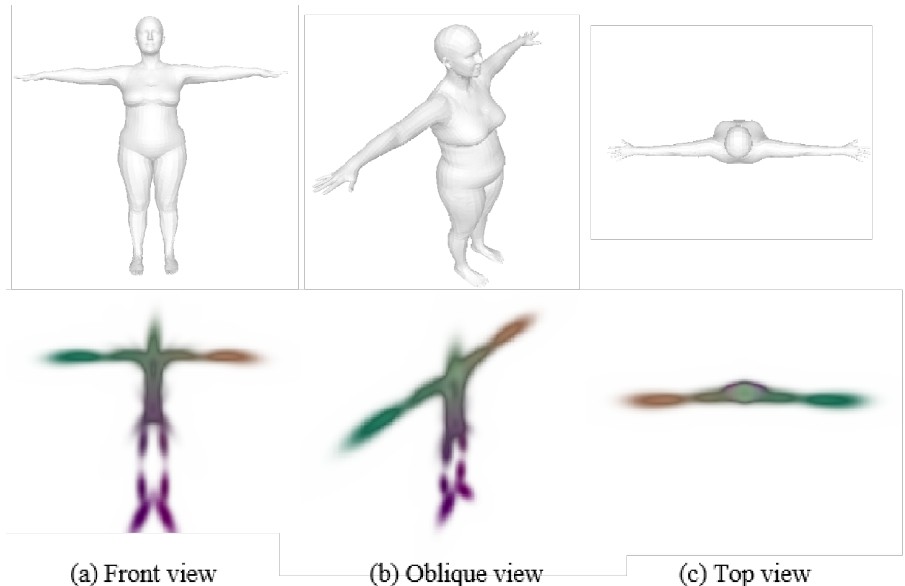

(a) Front view    (b) Oblique view    (c) Top view

Figure 6: Visualization of kernels for volumetric neural human from different views with a reference to the mesh model. Each body part is represented by one gaussian ellipsoids kernel.

where $\gamma$ is a coefficient that determines the rate of absorption, $t_n$ and $t_f$ donate the near and far bound along the ray.

Specifically, given volumetric human representation consists of $K$ rendering kernels, the human volume $\rho$ is reconstructed with a sum of these ellipsoidal Gaussians by the volume renderer:

$$\rho(X) = \sum_{k=1}^{K} \frac{1}{\sqrt{2\pi \cdot \|\textstyle\sum_k\|_2}} \, e^{-\frac{1}{2}(X-M_k)^T \cdot \sum_k^{-1} \cdot (X-M_k)} \tag{10}$$

where $K$ is the total number of Gaussian kernels of volumetric humans, $X = (x, y, z)$ is an arbitrary location in the 3D space. The $M_k$, a 3×1 vector, is the center of $k$-th ellipsoidal Gaussians kernel located on the human skeleton. The $\Sigma$ is a 3×3 spatial variance matrix. Specifically, by changing its scale and lineage, the direction, size and shape of $k$-th kernel can be fitted to the shape of the human body, as we show in A.3.1.

With the core parameter $\rho$ calculated from the volumetric human representation, the rendered feature maps $\overline{F}$ can be interpolated from the attribute of each reconstruction kernel via ray tracing, following the volume neural renderer Wang et al. (2022).

### A.4 Complete Derivations of the Formula

Here we develop derivations of the formula used in our analysis-by-synthesis pipeline for reasoning occlusion.

#### A.4.1 Derivation of Equation 1

Following Wang et al. (2021), we define a probabilistic generative models $p(F|\mathfrak{N}_y)$. Specifically, the likelihood of a target feature map $F \in \mathbb{R}^{H \times W \times D}$ is defined such that:

$$p(F|\mathfrak{N}, \Pi, B) = \prod_{i \in \mathcal{FG}} p(f_i|\mathfrak{N}) \prod_{i' \in \mathcal{BG}} p(f_{i'}|B), \tag{11}$$

where the foreground $\mathcal{FG}$ is the set of all positions on the 2D lattice $\mathcal{P}$ of the feature map $F$ that is covered by the rendered neural mesh model. The background $\mathcal{BG}$ contains those pixels respectively that are not covered by the mesh. The foreground likelihood $p(f_i|\mathfrak{N})$ is defined as a Gaussian distribution $\mathcal{N}(\phi_r, \sigma_r^2 I)$ with the mean vector $\phi_r$ and standard deviation $\sigma_r$. The correspondence between the image feature $f_i$ and the vertex feature $\phi_r$ is determined through the rendering process. Background features are modeled using a simple background model $p(f_{i'}|B)$ that is defined by a Gaussian distribution $\mathcal{N}(b, \sigma^2 I)$ with $B = \{b, \sigma\}$, which can be estimated with maximum likelihood from the background features.

To reason robustness, following related work on robust inference with generative models Egger et al. (2018), the model likelihood can be made robust to occlusion, as introduced in Section 2.2:

$$p(F|\mathfrak{N}, \Pi, B, z_i) = \prod_{i \in \mathcal{FG}} [p(f_i|\mathfrak{N})p(z_i{=}1)]^{z_i} [p(f_i|B)p(z_i{=}0)]^{(1-z_i)} \prod_{i' \in \mathcal{BG}} p(f_{i'}|B), \tag{12}$$

where $z_i \in \{0,1\}$ is a binary variable and $p(z_i{=}1)$ and $p(z_i{=}0)$ are the prior probabilities of the respective values. The variable $z_i$ allows the background model $p(f_i|B)$ to explain those locations in the feature map $F$ that is in the foreground region $\mathcal{FG}$, but which the foreground model $(f_i|\mathfrak{N})$ cannot explain well, presumably due to partial occlusion.

#### A.4.2 Derivation of Equation 2

Due to the correspondences between the feature vectors and the vertices are known, the training of the generative model parameters $\{\Pi, B\}$ and the feature extractor can be done by a simple maximum likelihood estimation (MLE) from the training data in our work. Also, during inference the human pose can be estimated by minimizing the negative log-likelihood of the model w.r.t. the camera parameters $\Pi$:

$$\mathcal{L}_{NLL}(F, \mathfrak{N}, \Pi, B) = -\ln p(F|\mathfrak{N}, \Pi, B) \tag{13}$$

$$= -\sum_{i \in \mathcal{FG}} \left( \ln\left(\frac{1}{\sigma_r \sqrt{2\pi}}\right) - \frac{1}{2\sigma_r^2} \|f_i - \phi_r\|^2 \right)$$

$$- \sum_{i' \in \mathcal{BG}} \left( \ln\left(\frac{1}{\sigma \sqrt{2\pi}}\right) - \frac{1}{2\sigma^2} \|f_{i'} - b\|^2 \right)$$

Assuming unit variances Wang et al. (2021), i.e. $\sigma^2 = \sigma_r = 1$, we reduce it to the mean squared error between the vertex features and the target feature map to use, as introduced in Section 2.2.

$$\mathcal{L}_{NLL}(F, \mathfrak{N}, \Pi, B) = \frac{1}{2} \sum_{i \in \mathcal{FG}} \|f_i - \phi_r\|^2 + \frac{1}{2} \sum_{i' \in \mathcal{BG}} \|f_{i'} - b\|^2 + C, \tag{14}$$

### A.5 Implementation Details

Here we develop more implementation details, including experimental setting and implementation.

**Training and Evaluation.** As a common practice, we train our volumetric neural human on **COCO** Lin et al. (2014) datasets for 175K iterations, i.e. 300 epochs. Since training requires the correspondence between image pixels and mesh vertices, ground truth SMPL parameters are needed. Since COCO is an in-the-wild dataset with only 2D annotations, we use pseudo-ground truth SMPL parameters generated by EFT Joo et al. (2021). We only use the training set of these datasets following

prior arts Kolotouros et al. (2019); Kocabas et al. (2021); Joo et al. (2021). Note that for all methods compared in the evaluation, we use the model trained with the *same* data as ours for fairness.

The test set **3DPW** von Marcard et al. (2018b) is used for which we sample the videos every 30 frames. Here, different from common practice, we also report the 2D Percentage of Correct Keypoints with head length threshold (PCKh) to measure how well the prediction aligns with the 2D image. Note that all methods are not trained on this dataset.

**Adversarial Occlusion Robustness Evaluation.** Inspired by the occlusion sensitivity analysis in Kocabas et al. (2021), we design an adversarial protocol **3DPW-AdvOcc** to evaluate the robustness of state-of-the-art (SotA) regression-based methods and how much our model can improve it. Specifically, we slide an occlusion patch over the input image to find the worst prediction the regressors make. This is done by comparing the relative performance degradation on the visible joints. We argue that evaluating the performance of occluded joints is sometimes ambiguous since the location of occluded joints is not always predictable even for a human. Therefore, for a more stable and meaningful evaluation, we only calculate the metrics on the joints that are not masked by the occlusion patch. Instead of using a gray occlusion patch, we use textured patches generated by randomly cropping texture maps from the Describable Textures Dataset (DTD) Cimpoi et al. (2014), which is more challenging. Two different patch sizes are used: $40 \times 40$ and $80 \times 80$ for a $224 \times 224$ image, denoted as Occ@40 and Occ@80 respectively.

**Experimental Implementation.** For training, We use a U-Net Ronneberger et al. (2015) style network as the feature extractor which consists of a ResNet-50 He et al. (2016) backbone and 3 upsampling blocks. The input image is a $224 \times 224$ crop centered around the human. The feature map has a resolution of $56 \times 56$ and the feature dimension is $128$. The Adam optimizer with a learning rate of $5 \times 10^{-5}$ and batch size of $64$ is used for training the feature extractor. Standard data augmentation techniques are used including random flipping, scaling and rotation. Note that unlike many regression-based methods Sárándi et al. (2018); Georgakis et al. (2020); Kocabas et al. (2021), we do not use occlusion augmentation. For inference, we use Adam as the optimizer with a learning rate of $0.02$ and run a maximum of $80$ steps. Different regressors are used as initialization. We check the negative log-likelihood $\mathcal{L}_{NLL}$ of the initial pose and its $180°$-rotated version around the y-axis and use the better one to initialize our model. We use the VoGE Wang et al. (2022) as our differentiable volume renderer.

