# OpenReview forum: "Volumetric Neural Human for Robust Pose Optimization via Analysis-by-synthesis"
_NeurIPS.cc/2022/Workshop/SVRHM — SVRHM Poster_

### Official Review · Reviewer_BveV · 2022-10-11
**Good and complete work but too densely compressed.**

**Rating:** 6
**Confidence:** 3

**Review:**

**Clarity:** This paper desires more clarity.

But I feel the lack of clarity majorly comes from the need to compress the paper into 4 page format, and many information are left to the appendix. This is a quite complete work that may fit well in a longer paper format for conferences like CVPR, and a longer 6-8 page format may help a lot on its clarity.

- **Position of figures**

    I hope the authors can move some figures to the main text to illustrate their model pipeline. The Figure 1 in appendix did a great job explaining and should come to front page in the camera ready version.

- **Notation of the system**

    In section 2.1, the paper says “a set of Gaussian ellipsoids kernels $V\in \mathbb R^{N\times 3}$ with feature vectors $\Phi=\{\phi_r\in \mathbb R^D\}$”. It’s not clear how many feature vectors are there. It’s also not clear what $N\times 3$ means for a Gaussian ellipsoid kernel. Is it the matrix consist of the center vector of each Gaussian kernel?  The meaning of $N$ and $D$ are also not described explicitly, which could only be inferred by the readers. Same issue for the subscript $r$ of $\phi_r$, not sure if it means “ray” or an index.

    In equation 1, the symbol $B$ is introduced without explanation. I think it’s the background scene of that image?

    In equation 1, $\mathcal F\mathcal G,\mathcal B\mathcal G$ are not described clearly, I feel it’s a set of index in the “feature map” plane. But how is it initialized? by ground truth location? What if you don’t know the foreground background segmentation from the start?

    The symbol $b$ in equation 2 is also not properly introduced. I can only assume it’s some feature vector of the background, but I’m not sure if it changes across space or not.

    Some of these information could be found in A.4.1, but I think it’s advisable make the notations clear in the main text, or you may lose your reader.

- The “**Adversarial Occlusion Robustness Evaluation” is not described adequately in the main text.**

    Specifically, the main text didn’t even describe the **“adversarial” part**. But the appendix description is better. If I understood it correctly, this evaluation may result in different positions of adversarial texture patches for different models? I just want to make sure the adversarial patch is not optimized to attack one of the system and then the same patch is used to evaluate other systems.


**Novelty/Originality**

- This works is a novel combination the idea of *analysis by synthesis*, *volumetric representation of human body* using Gaussian kernels, *contrastive feature learning.* It also leverages the recent development in differentiable volumetric rendering to help the analysis-by-synthesis optimization procedure.
- The authors have also showed its advantage in improving tackling adversarial occlusion problem, which is important in vision.

**Significance**:

- From the result evaluation table, it could be concluded that the proposed method have an advantage over existing systems in face of more severe , more adversarial occlusion. From the Figure 1 in appendix, it seems the analysis by synthesis system can qualitatively recover human pose in occluded scenario well.
- From the ablation study, we can also see the contributions of the several components of the system.

**Limitation:**

- **Inference efficiency comparison**

    Usually analysis by synthesis involves iterative optimization of the model and passing gradient through differentiable rendering pipeline. It will limit the computational efficiency of the system.

    So I’d like the authors to report the inference run time of their system and compare that to the previous regression based models. I can imagine there will be a difference. In Figure 5 in Appendix, they reported the training memory and speed, but didn’t report the inference time / memory.

- **Report of Computational Resources**

    Usually it’s good to report the computational resources used for the project and infrastructure they benchmarked the system on for training or inference.